# Characterization of Two Novel Single-Stranded RNA Viruses from *Agroathelia rolfsii*, the Causal Agent of Peanut Stem Rot

**DOI:** 10.3390/v16060854

**Published:** 2024-05-27

**Authors:** Dongyang Yu, Qianqian Wang, Wanduo Song, Yanping Kang, Yong Lei, Zhihui Wang, Yuning Chen, Dongxin Huai, Xin Wang, Boshou Liao, Liying Yan

**Affiliations:** Oil Crops Research Institute, Chinese Academy of Agricultural Sciences/Key Laboratory of Biology and Genetic Improvement of Oil Crops, Ministry of Agricultural and Rural Affairs, Wuhan 430062, China; yudongyang0730@aliyun.com (D.Y.); wangqianqian01@caas.cn (Q.W.); songwanduo@caas.cn (W.S.); kangyanping@caas.cn (Y.K.); leiyong@caas.cn (Y.L.); wangzhihui@caas.cn (Z.W.); chenyuning@caas.cn (Y.C.); dxhuai@163.com (D.H.); wangxin456_2000@163.com (X.W.)

**Keywords:** mitovirus, *Agroathelia rolfsii*, characterization, ArMV1, ArMV2

## Abstract

Peanut stem rot is a soil-borne disease caused by *Agroathelia rolfsii*. It occurs widely and seriously affects the peanut yield in most peanut-producing areas. The mycoviruses that induce the hypovirulence of some plant pathogenic fungi are potential resources for the biological control of fungal diseases in plants. Thus far, few mycoviruses have been found in *A. rolfsii*. In this study, two mitoviruses, namely, Agroathelia rolfsii mitovirus 1 (ArMV1) and Agroathelia rolfsii mitovirus 2 (ArMV2), were identified from the weakly virulent *A. rolfsii* strain GP3-1, and they were also found in other *A. rolfsii* isolates. High amounts of ArMV1 and ArMV2in the mycelium could reduce the virulence of *A. rolfsii* strains. This is the first report on the existence of mitoviruses in *A. rolfsii*. The results of this study may provide insights into the classification and evolution of mitoviruses in *A. rolfsii* and enable the exploration of the use of mycoviruses as biocontrol agents for the control of peanut stem rot.

## 1. Introduction

Mycoviruses, which are viruses that infect fungi, were first reported in diseased mushroom sporophores in 1962 [1], and were discovered in plant pathogenic fungi in 1971 [2]. They include ssRNA viruses, dsRNA viruses, and DNA viruses, and can be classified into 26 mycoviridae and 1 unclassified fungal virus genus. Most mycoviruses infect fungi latently and do not result in symptoms [3]. Some mycoviruses are beneficial to their hosts, increasing their virulence, sporulation, laccase activity, pigmentation [4], and pathogenicity [5]. Meanwhile, some mycoviruses reduce the virulence of their fungi hosts, such as Fusarium graminearum virus 1 (FgV1), Botrytis porri RNA virus 1 (BpRV1), and Sclerotinia sclerotiorum partitivirus 1 (SsPV1) [6,7,8].

Peanut (*Arachis hypogaea* L.) is an important oilseed and cash crop in China, with its production and yield ranking first among oilseed crops [9]. In the last twenty years, due to global warming, continuous cropping, and the return of straw to fields [10,11,12,13], peanut stem rot, caused by *Agroathelia rolfsii*, has become one of the most important soil-borne diseases in China, with the potential to induce 80% yield loss in fields that are seriously affected by the disease [14]. The utilization of resistant varieties is the most economical and effective measure used to control soil-borne diseases, but only a few peanut varieties with moderate resistance to stem rot have been identified so far [15,16,17,18]. The procedures used to control peanut stem rot mainly depend on chemical fungicides and agronomic practices [15], but these methods are not very efficient. To protect the environment and reduce the resistance of pathogenic fungi to fungicide, it is important to find an environmentally friendly and efficient means of managing the disease.

In recent years, with the development of mycovirus research, the hypovirulent mycoviruses of plant pathogenic fungi have been found to be an effective measure to control the host pathogen [19]. For example, Cryphonectria hypovirus 1 (CHV1) was used to efficiently control chestnut blight, with 70% of the cankers healed after 3 years of inoculation with the hypovirus [20,21]. Sclerotinia sclerotiorum hypovirulence-associated DNA virus 1 (SsHADV1), when used at the early flowering stage [22], was found to reduce both the incidence and severity of Sclerotinia stem rot by 21% and improve the oilseed yield by 13.7%. Regarding *A. rolfsii*, only a few reports on mycoviruses have been published since 2016. To date, a total of 22 mycoviruses have been identified from the *A. rolfsii* strains that infect Macleaya cordata [23]. Among them, Agroathelia rolfsii hypovirus 1 (SrHV1), Agroathelia rolfsii RNA virus 1 (SrRV1), and Agroathelia rolfsii mycovirus dsRNA 1 could reduce the virulence of *A. rolfsii*; these are thought to be hypoviruses [24]. However, no mycoviruses have yet been identified in *A. rolfsii*-infected peanut. This study is the first report on the identification of mycoviruses in the *A. rolfsii* strain GP3-1, which has a low virulence and was isolated from peanut. The results presented here might be useful for the exploration of mycoviruses as biocontrol agents for peanut stem rot.

## 2. Materials and Methods

### 2.1. Total RNA Extraction

The weakly virulent *A. rolfsii* strain GP3-1 was collected from peanut in 2013 [25,26]. It was cultured on potato dextrose agar (PDA) plates for 3–4 days at 30 °C in the dark, and subcultured on PDA overlaid with 0.02 mm cellophane films. Mycelium was collected and ground into powder in liquid nitrogen, then stored at −80 °C. The total RNA of *A. rolfsii* GP3-1 was isolated from approximately 100 mg of mycelial powder using TRIzol (NEWBIOINDUSTRY, Tianjin, China), following the manufacture’s protocol; it was then treated with DNase I (TaKaRa, Dalian, China). The quality of the purified RNA was assessed using 1% agarose gel electrophoresis, and its quantity was measured using Nanodrop (Thermo Fisher Scientific: Waltham, MA, USA).

### 2.2. RNA High-Throughput Sequencing

Metagenomic next-generation sequencing (mNGS) has been employed to identify mycoviruses in *A. rolfsii* GP3-1. A total of 2 µg of purified RNA was sequenced by Illumina Hiseq 2500. Clean reads were obtained by removing adaptors and low-quality reads, and then mapping them onto the reference genome of *A. rolfsii* GP3-1 [26]. To avoid false-positives, sequence similarity cut-off values of 1E-5 and 1E-10 were used for the Nucleotide Sequence Database (NT) and Non-Redundant Protein Sequence Database databases (NR), respectively. Non-host reads were assembled using the splicing software Trinity in order to obtain contigs, and then the contigs were blasted in the GenBank Virus RefSeq protein database using DIAMOND software [27] to obtain the potential viruses (default parameter). The assembled contigs were also blasted in the NR, NT, and CDD databases of the NCBI to search for putative viruses.

### 2.3. RT-PCR and Rapid Amplification of cDNA Ends (RACE)

By using the total RNA of the strain GP3-1 as a template, cDNA was synthesized us-ing a reverse transcription kit (Thermo, MA, USA) with a random primer. Viral fragments were amplified with cDNA and specific primers (Appendix A), and cloned into a pEASY vector. The 5′ and 3′ end of the putative mitoviruses were determined via the rapid amplification of the cDNA end (RACE) [28]. Approximately 1 µg of the total RNA of GP3-1 was li-gated with the linker RACE-OLOGO using T4-RNA ligase (Thermo, Shanghai, China). The ligated RNA sample was then reverse-transcribed into cDNA with the primer RACE1. For 5′ end and 3′ end of the virus, the first PCR was amplified using RACE2 and Ra1-R2 or Ra1-F2, respectively. The second PCR that was amplified used the product of the first PCR as a template, employing RACE3 and Ra1-R1 or Ra1-F1, respectively. Then, the second PCR products of 5′RACE and 3′RACE were purified with a gel extraction kit (Omega Bio, GA, USA), and then cloned with a pEASY vector (TRANS, Beijing, China) for sequencing. The specific sequences of all primers are listed in Appendix A. The experiment was repeated twice.

### 2.4. Sequence Analysis

The open reading frames (ORFs) of the viruses were predicted using the ORF finder (https://www.ncbi.nlm.nih.gov/orffinder/ accessed on 3 November 2023) program. The conserved motif of the amino acid (aa) sequences of the putative mycoviruses were determined via a search of the Conserved Domain Database (CDD) (with e-value of 0.01) (http://www.ncbi.nlm.nih.gov/Structure/cdd/wrpsb.cgi/ accessed on 3 November 2023), Protein Family (Pfam) database (http://pfam.sanger.ac.uk/ accessed on 3 November 2023), and PROSITE database (http://www.expasy.ch/ accessed on 4 November 2023). The stem–loop structures in the 5’ and 3’ UTRs were predicted using RNAfold (http://rna.tbi.univ.ac.at//cgi-bin/RNAWebSuite/RNAfold.cgi/ accessed on 4 November 2023). The nucleotide sequence alignment of the mitoviruses was performed using Clustalx 2.1. Phylogenetic trees of mitoviruses from *A. rolfsii* were constructed with mitoviruses from other hosts (Appendix A); this was achieved using IQ-Tree and the maximum likelihood (ML) method in a VT + F + I + R6 model with 1000 bootstrap replicates [29,30,31,32]. The multiple comparison diagrams of the conserved domains were created using GenDoc. Viral schematic diagrams were created using Software IBS 2.0 (https://ibs.renlab.org/ accessed on 4 November 2023) [33].

### 2.5. Determination of Mitovirus in Other A. rolfsii Strains

To detect the host range of those mitoviruses, total RNA was purified from other *A. rolfsii* strains in the same mycelial compatibility group (MCGs) as GP3-1, including BL1-1, BL1-2, GP1-1, GP1-2, FC1-2, FC2-1, FC2-2, and GZ [34], and *A. rolfsii* strains from other MCGs (BS1, BSH1, FJT, GZH1, HN1, KL1, JZ1, SX1, TSH1, XJ1, YD1, YL1, ZY2, and CHJ1) (Appendix A). The viral specific primers 1-R2 or 2-R2 were used to synthesize the first-strand cDNA of mitovirus 1 and mitovirus 2, respectively; then, the specific fragment of putative mitovirus 1 was amplified with 1-F2 and 1-R2, and the specific fragment of putative mitovirus 2 was amplified with 2-F2 and 2-R2. For each semi-quantitative PCR amplification, the tubulin gene of GP3-1 was used as an internal reference. The 20 μL reaction system of Semi-QRT-PCR contained 2 × PrimeSTAR Max Premix 10 μL, forward and reverse primers 1 μL, and template cDNA 100 ng, and we add ddH2O to 20 μL, respectively. The program is: 98 °C 3 min, 34 cycles (98 °C 10 min, 60 °C 15 s, and 72 °C 1 min), and 72 °C 3 min. The 5 μL PCR product and Marker were detected on 1% agarose gel.

### 2.6. Effect of Mycovirus on A. rolfsii Pathogenicity

We detected the mycelial fusion of GP3-1 with the moderately virulent strain BL1-1 in the same MCG: mycelial disks of GP3-1 and BL1-1 were placed on the same PDA medium at a distance of 3 cm and cultured at 30 °C in the dark. When the mycelia of two strains made contact, the mycelia that made contact at a distance of 1 cm were punched and transferred to a fresh PDA plate; these were named BG1. The mycelial disks of GP3-1 and BG1 were also cultured on the same PDA plate, and the disk at the place of contact was obtained and named BG2. This was repeated until the tenth generation of the mycelial fusion clone BG10 was obtained. The total RNA of the mycelia of GP3-1, BL1-1, BG5, and BG10 was extracted. ArMV1 and ArMV2 in those four strains were identified by qRT-PCR using the specific primers of each virus. The tubulin gene of *A. rolfsii* was used as an internal reference. The qRT-PCR system contains 10 μL 2 × ChamQ Universal SYBR qPCR Master Mix, 0.5 μL of upstream and downstream primers, 2 μL template DNA, and 7 μL H_2_O, respectively. The procedure is as follows: 95 °C 1 min, 40 cycle is 95 °C 10 s, and 60 °C 30 s, and the dissolution curve is 95 °C 15 s, 60 °C 60 s, and 95 °C 1 s. The relative expression of the gene was calculated by 2^−ΔΔCT^ algorithm.

In order to determine whether genetic background affects the pathogenicity of BG5 and BG10, the PCR was amplified with the cDNA of GP3-1, BL1-1, and BG10, the mixed DNA of the GP3-1 and BL1-1 templates, respectively, and the specific primers GCE1-F, GCE1-R, GCE2-F, and GCE2-R (Appendix A).

The pathogenicity of clone BG5 and BG10 was tested on peanut leaves. The peanut variety Zhonghua 9 was planted in the field for 30 days, and then the reciprocal third leaves from the main stems were taken and put on a wet paper towel in the petri dishes. The mycelial disks of GP3-1, BL1-1, BG5, and BG10 were separately placed on the leaves with the mycelium side down, and cultured at 30 °C in the dark. The lesion diameter was measured using the cross method, 48 h after inoculation. Each strain was inoculated on four leaves and the experiment was repeated three times. GraphPad Prism 8.0.2 was used for statistical data and mapping.

## 3. Results

### 3.1. Characteristics of ArMV1 and ArMV2 Genome

We obtained 45,409,042 raw reads in GP3-1. The Q20, Q30, and GC content were 97.11%, 91.76%, and 42.75% respectively. Once the low-quality reads had been removed, a total of 40,006,732 non-host reads were obtained by RNA sequencing. After being blasted in the protein, conserved domain, and nucleotide GenBank databases, two sequences (contig1 and contig2) matched the mitovirus. Contig1 and contig2 shared the highest level of amino acid identity with Rhizoctonia solani mitovirus 115, at 36.27% and 35.12%, respectively (Appendix A). Meanwhile, contig1 and contig2 shared less than 1% nucleotide sequence identity with any other reported viruses. Via a comparison of the sequences between contig1 and contig2, we found that contig-1 shared 20.63% of amino acid identity and 46.8% of nucleotide sequence identity with contig2, respectively (Table 1). Thus, contig1 and contig2 were considered two novel species in the genus mitovirus. We named contig1 as Agroathelia rolfsii mitovirus 1 (ArMV1) and contig2 as Agroathelia rolfsii mitovirus 2 (ArMV2). The metatranscriptomic sequencing data of GP3-1 have been deposited in the Sequence Read Archive (SRA) with the following accession numbers: SAMN38757371.

The whole genomes of these two putative mitoviruses of *A. rolfsii* GP3-1 were cloned using RT-PCR, 5′ RACE, and 3′RACE. The complete genome size of ArMV1 was 3, with 514 nucleotides (nt) and a GC content of 37% (Table 1). It contained one large open reading frame (ORF) of 2781 nt encoding a polyprotein RNA-dependent RNA polymerase (RdRp) of 927 amino acids. Its 5′ untranslated region (UTR) had 564 nt and its 3′ UTR had 169 nt (Figure 1A). The complete genome sequence of ArMV2 was 3410 nt with a GC content of 39%. It encoded one large ORF that putatively encoded an RdRp of 937 amino acids. Its 5′UTR had 365 nt and 3′UTR had 234 nt (Figure 1B). The percentage of the UGA and UGG codon of tryptophan was 8.3% and 91.7% in ArMV1, and 0% and 100% in ArMV2, respectively (Table 1). The sequence of ArMV1 and ArMV2 was submitted to the GenBank database with the accession numbers OR995730 and OR995731, respectively.

The amino acid sequence alignment of ArMV1 and ArMV1 with those mitoviruses from other fungal hosts (Appendix A) indicated that there were seven highly conserved motifs (motif I to motif VII) in the RdRp protein of mitoviruses, as reported by previous studies; these include motif II (TFNQ), motif III (DLS and ATD), motif IV (GQP), motif V (GDD), and motif VII (EFAK) (Figure 2).

### 3.2. Phylogenetic Analysis of ArMV1 and ArMV2 with Other Mitoviruses

A phylogenetic tree was constructed with IQ-Tree and Clustal X using the putative RdRp amino acid sequence of ArMV1 and ArMV2 and forty others fungal mitoviruses (Appendix A); these mitoviruses were then assigned into five clusters, with cluster I~VI including species *Duamitovirus, Triamitovirus, Unuamitovirus,* and *Kvaramitovirus*, respectively. Although ArMV1 and ArMV2 did not belong to any clusters, they were closely related to the mitovirus obtained from *Rhizoctonia solani* (Figure 3).

### 3.3. Secondary Structure Prediction

The stem–loop structures of the 5’-UTR and 3’-UTR of ArMV1 and ArMV2 were predicted using RNAfold. It was predicted that the 5′ terminus (1–45 nt) and the 3′ terminus (3365–3514 nt) of ArMV1 would fold into stem–loop structures, with ΔG values of −4.12 kcal/mol and −11.74 kcal/mol, respectively. The stem–loop structure of the 5ʹ-UTR (1–37 nt) and 3ʹ-UTR (3365–3410 nt) of ArMV2 had initial ∆G values of −7.87 kcal/mol and −6.09 kcal/mol, respectively. In addition, it was predicted that the 5′ UTR (1−16 nt) and 3′−UTR (3494−3508 nt) of ArMV1 and the 5′ UTR (1−10 nt) and 3′ UTR (3399−3406 nt) of ArMV2 would generate a potential panhandle structure with initial ∆G values of −6.47 kcal/mol and −5.35 kcal/mol, respectively (Figure 4).

### 3.4. Detection of ArMV1 and ArMV2 in Other A. rolfsii Strains

To determine whether ArMV1 and ArMV2 existed in the other nine *A. rolfsii* strains belonging to the same MCG as GP3-1, we used RT-PCR to amplify the cDNA of these strains with a specific primer of ArMV1 and ArMV2; we also detected ArMV1 and ArMV2 in the other 14 *A. rolfsii* strains belonging to other MCGs (each strain per MCG). The result showed that 10 strains from other MCGs contained both viruses; however, strain FJT from Jilin province only carried ArMV2, strain SX1 from Anhui province only carried ArMV1, and strain BSH1 from Fujian province and strain TSH1 from Liaoning province did not contain either ArMV1 or ArMV2 (Table 2 and Figure 5).

The results of the semi-quantitative PCR showed that all these strains in the same MCGs as GP3-1 also contained ArMV1 and ArMV2, but that the amount of ArMV1 and ArMV2 in these strains was lower than that in GP3-1 (Figure 6); in addition, the amount of ArMV1 and ArMV2 in BL1-1 was the lowest among the strains that carried both viruses. By comparing the amount of ArMV1 and ArMV2 in GP3-1 with the 10 strains from other MCGs that contained both viruses, we found that the amount of ArMV1 and ArMV2 in these strains was lower than that in GP3-1. The amount of ArMV1 and ArMV2 in ZY2 was the lowest among the 10 strains that contained both viruses.

### 3.5. High Level of ArMV1 and ArMV2 Accumulation Affects the Pathogenicity of A. rolfsii Strains

The diameter of the lesions that appeared on peanut leaves inoculated with GP3-1, BL1-1, and BG5 and BG 10 (two fusion clones of GP3-1 and BL1-1) was compared. Com-pared to the BL1-1 wild strain, BG5 and BG10 produced smaller lesions; meanwhile, the amount of ArMV1 and ArMV2 was higher in BG5 and BG10 compared to BL1-1. There was a significant difference in the viral load amount observed between BL1-1 and BG5, and between BL1-1 and BG10 (Figure 7). The results showed that there was a significant difference in the lesion diameter between BL1-1 and BG5, between BL1-1 and BG10, and between BG5 and BG10 (Figure 8). The lesion diameter showed a significant negative cor-relation with the amount of ArMV1 and ArMV2, with correlation efficiencies of −0.981 and −0.967 (Table 3). Based on the PCR amplified, to understand the genome of the fusion clone BG10 being similar with BL1-1 or GP3-1, the PCR reaction was amplified with the cDNA of GP3-1, BL1-1, and BG10, and the mixed DNA of GP3-1, and the BL1-1, respectively, and templates of with the specific primers GCE1-F, GCE1-R, GCE2-F, and GCE2-R; those primers could distinguish the genome of GP3-1 and BL1-1. From the PCR results, we found that the size of the amplified fragment of BG10 was the same as BL1-1, but not GP3-1, which was only affected by the genetic background of BL1-1, not GP3-1 (Figure 9).

## 4. Discussion

In the present study, we characterized two novel mycoviruses from the hypovirulent *A. rolfsii* strain GP3-1 using viral metagenomics, RT-PCR and RACE PCR. We found that these two viruses were new species of mycoviruses. As far as we know, the mycoviruses found in *A. rolfsii* belong to the families of *Enyviridae, Endornaviridae, Fusariviridae, Hypoviridae*, and *Fusagraviridae,* and unclassified viruses [23]. ArMV1 and ArMV2 were the first mitoviruses reported to belong to the *Mitoviridae* family that is found in *A. rolfsii*.

Mitoviridae is a family that was first reported by ICTV in 2021 [35]. ArMV1 and ArMV2 are two new members of the family *Mitoviridae*, with +ssRNA and encoding a large RdRp. The lengths of the RdRp sequences of ArMv1 and ArMV2 are 927 aa and 937 aa, respectively, which is longer than the length of the RdRp in most mitoviruses (600 aa~900 aa). The 5′-UTR of ArMV1 was 564 nt in length, which is longer than that of most other mitoviruses, except that of Cryphonectria cubensis mitovirus 2a (892 nt) and Tuber excavatum mitovirus (793 nt) [36,37]. The 3′ UTR of ArMV1 and ArMV2 was 175 nt and 234 nt, respectively, which is shorter than that of most previously reported mitoviruses, except RsMV40 (20 nt) [37].

For viruses in the mitochondria, the codons of mitoviruses have evolved to adapt to those of the mitochondria. The UGA (stop codon in the standard genetic code) and UGG codons in mitoviruses encoded tryptophan (Trp) [38]. Based on previous reports, mitoviruses typically contain 0–20 UGA, and most of them are 7–12 UGA. In this study, there were 16 UGG codons in the ORF of ArMV1, and 14 UGG codons in the ORF of ArMV2. The percentages of UGA and UGG encoded for Trp in mitoviruses were 67% and 33% [38], while the percentages of UGA and UGG in ArMV1 and ArMV2 were 8.3% and 91.7% and 0% and 100%, respectively. ArMV1 and ArMV2 contained a lower percentage of UGA codons compared to other mitoviruses. The low number of UGA codons in these mitoviruses might be related to their host, as there is some evidence that the UGA (Trp) codon is rarely observable in the endogenous core mitochondrial genes of some fungi [38,39]. This shows that the ArMV1 and ArMV2 found in *A. rolfsii* may have evolved together with the mitoviruses of their fungal hosts.

The phylogenetic analysis performed in the present study indicated that mitoviruses are grouped into five clusters. ArMV1 and ArMV2 were in Cluster II, and were closely related to Rhizoctonia solani mitovirus 115 (RsMV115) and RsMV11. Mitoviruses from the same plant pathogenic fungus were grouped into different groups, while those from different hosts were closely related [40,41,42]. Some mitoviruses may have existed before their fungal hosts diverged [31]. ArMV1 and ArMV2 were found in *A. rolfsii* strains in the same MCG and in different MCGs, suggesting that ArMV1 and ArMV2 originated earlier than *A. rolfsii*.

The predicted secondary structures of the 5′-UTR and 3′-UTR of the ArMV1 and ArMV2 genome were similar to those of previously reported mitoviruses [32,43,44,45]; this suggests that the stem–loops and panhandle structures in the UTRs of mitoviruses might play an important role in their adaption to the host. Panhandle structures could be selectively recognized by the RdRp, which triggers both transcription to mRNA and replication into the complementary RNA in Influenza A virus (IAV) [46]. Panhandle structures might also play an important role in the transcription of the RdRp in mitoviruses.

Some mitoviruses, such as Sclerotinia sclerotiorum mitovirus 1 ~ 4 (SsMV1 ~ 4), have been identified as hypovirulent viruses [40,47]. The number amount of mycoviruses in pathogenic fungi affects the phenotype of the host fungi, and a high number amount of mycoviruses would reduce the virulence of the host; however, a low number amount of mycoviruses has little effect on the virulence of the host [48]. The 2- to 5-fold overexpression of dsRNA in the *A. rolfsii* strain PH-1 could cause different degrees of hypovirulence [49]. In this study, the virulence of GP3-1 was lower than that of the strains belonging to the same MCG and the strains in other MCGs [25,26]; meanwhile, the quantity of ArMV1 and ArMV2 in GP3-1 was higher than that of other strains. Furthermore, the results of the virus transmission experiment also showed that these strains contained a higher amount of ArMV1, and ArMV2 showed a lower level of virulence. The higher quantity of ArMV1 and ArMV2 in their host fungi may explain the lower virulence of GP3-1. In order to remove the mitoviruses in *A. rolfsii* GP3-1, we evaluated the efficacy of three different methods, including ribavirin with hyphal tip isolation, cycloheximide with hyphal tip isolation, and a high-temperature treatment with hyphal tip isolation. However, after two successive treatments, the quantity of ArMV1 and ArMV2 was the same as that in the wild host strain. These results are shown in Appendix A. It was suggested that ArMV1 and ArMV2 were not easy to remove from their host. The effects of mitoviruses on their host need to be studied further.

The present study constitutes the first report on mitoviruses in *A. rolfsii*, considering that the two viruses are present in many *A. rolfsii* strains from the same MCG and from other MCGs. This might provide a basis for exploring the interaction between and evolution of the virus and fungal hosts of different genotypes, and also provide insight into evaluating the potential application of mycoviruses as an agent for the biocontrol of *A. rolfsii*.

## Figures and Tables

**Figure 1 viruses-16-00854-f001:**
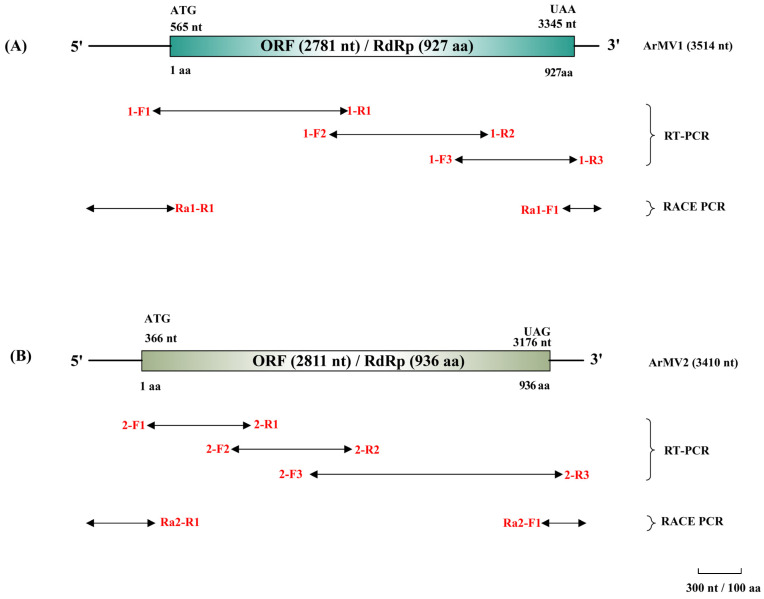
Diagrammatic sketch of RT-PCR and RACE of ArMV1 and ArMV2. (**A**) The genome organization and the polyprotein encoded by the ORF of ArMV1. (**B**) The genome organization and the polyprotein encoded by the ORF of ArMV2. The lines with a double arrow are the RT-PCR and RACE PCR products. The texts written in red show the names of the primers used.

**Figure 2 viruses-16-00854-f002:**
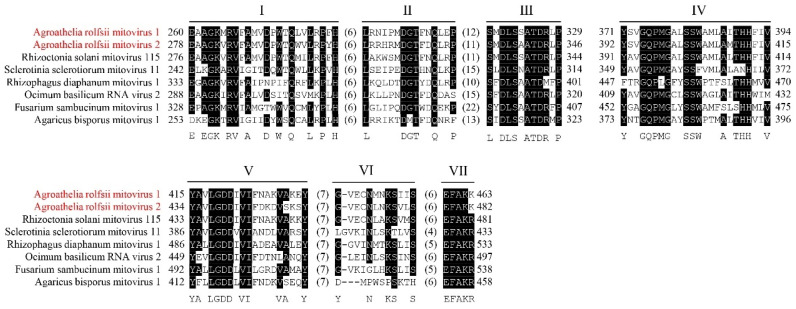
Alignment of the RdRp of selected viruses in *Mitoviridae*. ArMV1 and ArMV2 are shown in red. I~VII are the conserved domains.

**Figure 3 viruses-16-00854-f003:**
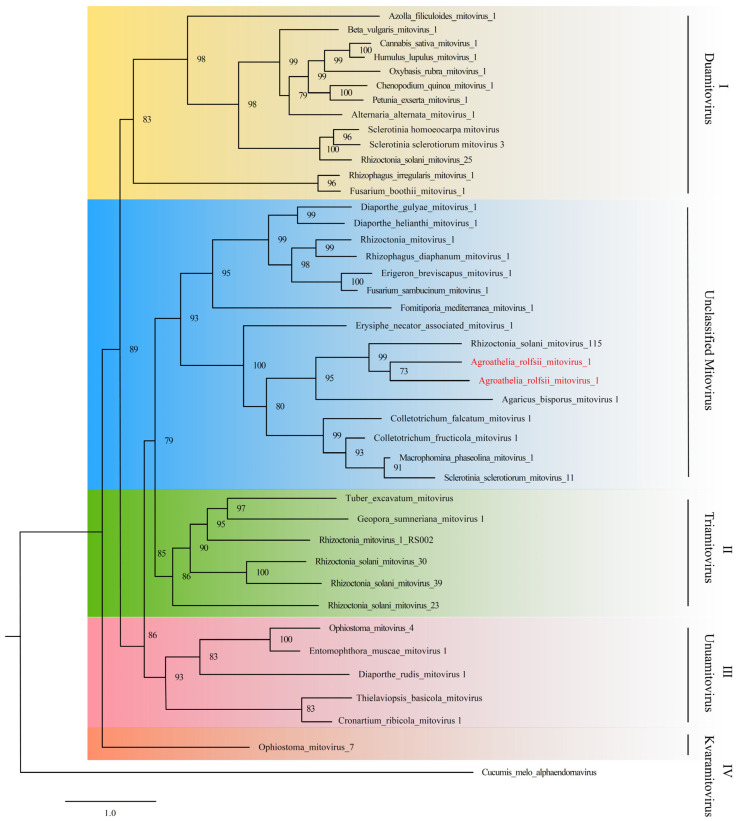
Phylogenetic analysis of ArMV1 and ArMV2 and other viruses in *Mitoviridae* based on the RdRP domain using the Maximum Likelihood program with 1000 bootstrap replicates. The text written in red shows the newly identified viruses.

**Figure 4 viruses-16-00854-f004:**
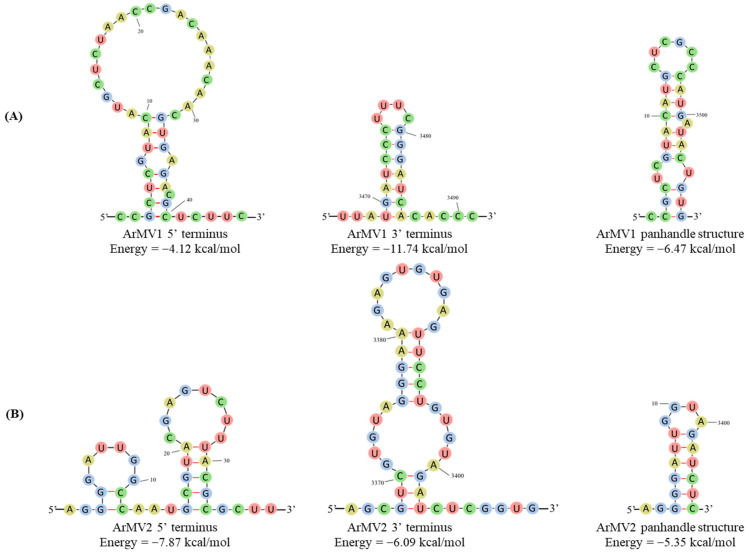
Predicted secondary structure: (**A**) the 5’ and 3’ termini and panhandle structures of ArMV1; and (**B**) the 5’ and 3’ termini and panhandle structures of ArMV2.

**Figure 5 viruses-16-00854-f005:**
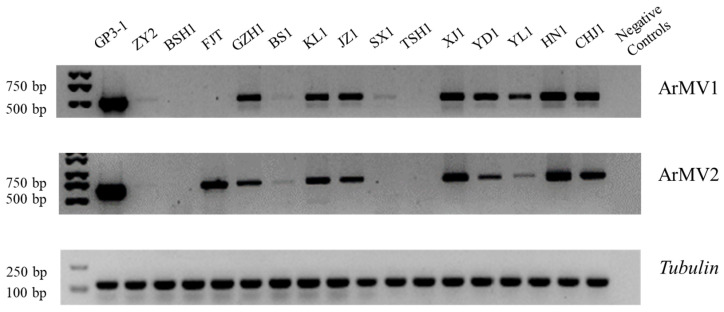
The results of the identification of ArMV1 and ArMV2 in the strains (Appendix A) obtained from 14 provinces in China. The maker used was the DL2000 maker.

**Figure 6 viruses-16-00854-f006:**
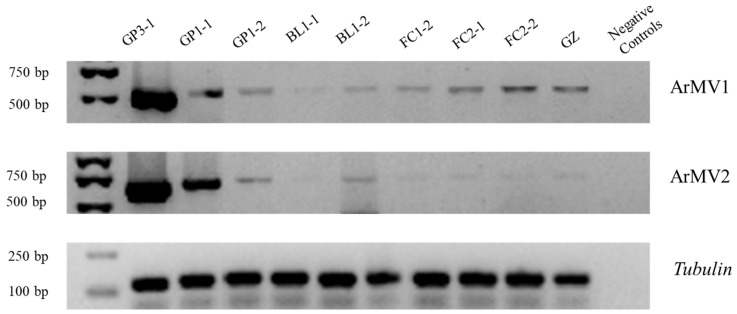
ArMV1 and ArMV2 were identified in the strains belonging to the same MCG as GP3-1. The maker used was the DL2000 maker.

**Figure 7 viruses-16-00854-f007:**
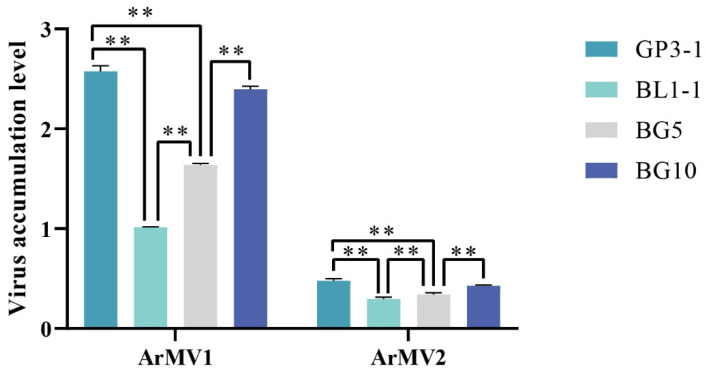
The difference in the level of ArMV1 and ArMV2 accumulation in GP3-1, BL1-1, BG5, and BG10. ** denotes *p* < 0.01.

**Figure 8 viruses-16-00854-f008:**
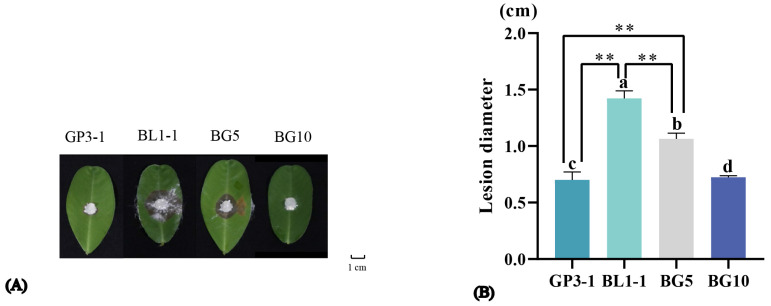
(**A**) Disease spots of Zhonghua 9 inoculated with GP3-1, BL1-1, BG5, and BG10 for 48 h. (**B**) The difference in the spot diameter of peanut infected by GP3-1, BL1-1, BG5, and BG10. ** denotes *p* < 0.01. a, b, c and d represents the lesion diameter of GP3-1, BL1-1, BG5 and BG10 respectively.

**Figure 9 viruses-16-00854-f009:**
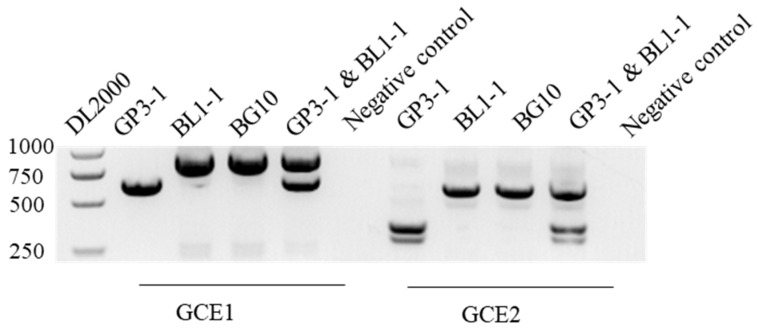
Identification of genetic background using BG10. Lane DL2000, DNA maker; Lane 1 (620 bp), the GCE1 of GP3-1; Lane 2 (832 bp), the GCE1 of BL1-1; Lane 3 (832 bp), the GCE1 of BG10; Lane 4 (832 bp and 620 bp), the GCE1 of GP3-1 and BL1-1; Lane 5 (340 bp), the GCE2 of GP3-1; Lane 6 (577 bp), the GCE2 of BL1-1; Lane 7 (577 bp), the GCE2 of BG10; and Lane 8 (577 bp and 340 bp), the GCE1 of GP3-1 and BL1-1.

**Table 1 viruses-16-00854-t001:** Nucleotide identity and information about ArMV1 and ArMV2.

	Nucleotide Sequence Identity (%)	Amino Acid Sequence Identity (%)	GC Content (%)	UGA Number *	UGG Number *	Trp = UGA (%)	Trp = UGG (%)
ArMV1	46.8	20.63	37	1	11	8.3	91.7
ArMV2			39	0	14	0	100

Note: * denotes Trp codon number in RdRp.

**Table 2 viruses-16-00854-t002:** Results regarding the origin and identification of ArMV1 and ArMV2 in 14 strains.

Strain Source (Province)	Mycelial Compatibility Groups (MCGs)	Isolate	Virus	
			ArMV1	ArMV2
Chongqing	MCG1	BS1	+	+
Fujian	MCG2	BSH1	-	-
Jiangxi	MCG3	CHJ1	+	+
Jilin	MCG4	FJT	-	+
Guizhou	MCG5	GZH1	+	+
Hainan	MCG6	HN1	+	+
Guangdong	MCG7	KL1	+	+
Shandong	MCG8	JZ1	+	+
Anhui	MCG9	SX1	+	-
Liaoning	MCG10	TSH1	-	-
Hebei	MCG11	XJ1	+	+
Sichuan	MCG12	YD1	+	+
Huibei	MCG13	YL1	+	+
Henan	MCG14	ZY2	+	+

Note: + denotes virus-positive; - denotes virus-negative.

**Table 3 viruses-16-00854-t003:** Correlation analysis of viral amount and diameter of lesions on the leaves.

Mycovirus	Strain	Virus Amount	Lesion Diameter (cm)	Significance
	GP3-1	2.568 ± 0.046	0.7 ± 0.03	−0.981 **
ArMV1	BL1-1	1.014 ± 0.005	1.4 ± 0.06	
	BG5	1.638 ± 0.012	1.1 ± 0.05	
	BG10	2.405 ± 0.032	0.8 ± 0.04	
	GP3-1	0.465 ± 0.024	0.7 ± 0.03	−0.967 **
ArMV2	BL1-1	0.299 ± 0.0179	1.4 ± 0.06	
	BG5	0.342 ± 0.017	1.1 ± 0.05	
	BG10	0.480 ± 0.018	0.8 ± 0.04	

Note: ** denotes *p* < 0.01.

## Data Availability

The original contributions presented in the study are included in the article, further inquiries can be directed to the corresponding author.

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
