# Peer review of "Characterization of Two Novel Single-Stranded RNA Viruses from Agroathelia rolfsii, the Causal Agent of Peanut Stem Rot"

_viruses, 2024, doi:10.3390/v16060854_

Round 1

Reviewer 1 Report

Comments and Suggestions for Authors

The manuscript described two novel single-stranded RNA viruses from Agroathelia rolfsii and their potential as biocontrol agents for peanut stem rot disease. However, some comments should be clarified before accepted for publication. Please see the comment in the attached file.

Comments on the Quality of English Language

 The manuscript is well-written and formatted.  However, it should be checked carefully before being accepted for publication. 

Author Response

"Please see the attachment

Reviewer 2 Report

Comments and Suggestions for Authors

Two novel mitoviruses from the fungus A. rolfsii have been characterised for the first time in this simple sequence-based report. I recommend that authors send it to any professional English editing service for corrections. There are also several careless typos in this manuscript. Furthermore, I am dissatisfied with the experiment design and representation of results. I suggest therefore that this paper undergo a major revision before it is accepted for publication.

Comments:

Line 18: Spelling correction for “Ccumulation”.

Line 28: There are other viral families reported from fungi as well. For example, Mitoviridae, Yadokariviridae, Genomoviridae etc. Please perform a careful review and modify the text accordingly.

L57: What is the difference between A. rolfsii that infects peanut and Macleaya cordata? Do they belong to different somatic incompatibility groups?

L67: Please mention what type of cellophane was used. Please mention specifications.

L74: Provide more information regarding RNA sequencing. For example, sequencing platform used, sample prep method, total reads generated, assembly criteria, no of assembled reads, read coverage etc.

L121: Why did you collect mycelial disk from the connected regions? It should be taken far from the fusion region on BL1-1 side to avoid any hyphal contamination from GP3-1 side. How did you distinguish between two different strains (donor and recipient)? How did you make sure that the resulting strain (BG5) is not the mixture of both the strains GP3-1 and BL1-1.

L136: Did you find any other fungal viruses in GP3-1?

L208: It should be ArMV1 not SrMV1 right?

L210: Where is the material and method for semi quantitative PCR and quantification of viral load in each strain?

L226: The effect of individual mitoviruses can be known by sequential removal of both viruses from their original host. How did you confirm that BL1-1 was a virus-free strain? Please provide colony morphology photographs for GP3-1, BL1-1, and BG5 to BG 10 side by side. It's possible that BL1-1 had a virus that was actually linked to its enhanced virulence, but when you generated BG series, that virus got eliminated, resulting in BL1-1 becoming less virulent. Here, you need to prove that BG5-BG10 are derivatives of BL1-1 (same genetic background), BL1-1 was a complete virus-free strain, and after hyphal anastomosis with GP3-1, both the mitoviruses were transferred to BL1-1 side generating BG5-10. The primary goal is to demonstrate that there is no hyphal contamination in the BG series from the GP3-1 strain. In this case, I think, the best way of doing this is to eliminate these mitoviruses from GP3-1 and then compare virulence, colony morphology etc. between virus-infected and virus-free GP3-1 strains. Figure 6 indicates presence of faint bands for both the mitoviruses in BL1-1. Does this mean that BL1-1 was not virus-free?

Figure 5: Why did you mention SrMV1 and SrMV2 right beside the figure? Is it a typo?

Figure 7: What was the method used to measure virus accumulation here? The accumulation levels of ArMV-1 and ArMV-2 are almost the same in GP3-1 (Figures 5 and 6). However, the results presented in Figure 7 are inconsistent with the results presented in Figure 5 and 6.

Table 3: Please mention units for virus accumulation and lesion diameter measurements.

L249: New species of mycoviruses??

L251: “ArMV1 and ArMV2 were the first reported mitoviruses belonged the family of Metaviridae found in A. rolfsii”. Is this a right statement? Metaviridae is a family of retrotransposons and reverse-transcribing viruses with long terminal repeats. As far as I am concerned mitoviruses don’t belong to this family. 

L278: I am not in agreement with this explanation. Although two strains are somatically incompatible, viruses can still be transmitted between them. There have been reports published on this before. It is also possible that A. rolfsii strains picked up both viruses from the same source. It is also possible that a plant host was originally a carrier of these viruses and was picked up by both fungal strains during different infection events.

Line 302: You should also try single spore culture technique for removal of mitoviruses from GP3-1

Comments on the Quality of English Language

The paper has lot of typos and English issues, so I recommend that authors send it to any professional English editing service for correction before they resubmit it for peer review.

Reviewer 3 Report

Comments and Suggestions for Authors

The work describes the characterization of two mitoviruses infecting the fungus Agroathelia rolfsii, causing agent of peanut stem rot. Two previous works describe the mycovirome of this fungus infecting other host, however, no previous mitoviruses have been identified associated with this plant pathogenic fungus. The manuscript has interesting data but a major revision should be performed before its publication. In general, the authors should review the English language, rewrite sentences that are not completely clear, check spaces between words, and review references to assure that are used in the right paragraph in the text.

Introduction: Check the review DOI: 10.1016/bs.aivir.2023.02.002 to correct family classification of mycoviruses, some families are missing. Reorganize the introduction to include the information about the effect of mycoviruses on the fungal hosts in the same paragraph.

According to the ICTV, Mitoviridae family has only four genera Duamitovirus, Kvaramitovirus, Triamitovirus, and Unuamitovirus. The authors sometimes talk about Metaviridae family (this is a family of reverse-transcribing viruses) when they refer to Mitoviridae family, that includes the new identified mitoviruses.

Materials and methods:

-Include the information about the attempts performed to cure fungal strain GP3-1 of mitoviruses.

-It is not clear how RACE was performed with a protocol designed for dsRNA mycoviruses using total RNA. Clarify this point.

-In point 2.6. it will be convenient to include a scheme of the process. It is not clear why the authors follow this protocol, what is the purpose to do several anastomosis events. Besides the strain used for transference of mitoviruses is already infected with both mitoviruses and cannot be used to determine their effect on virulence.

-In 2.5. change “Mitocovirus” by “mitovirus”

Results:

-Figures 2 and 3 are blurry.

-Table 1 correct “NucleNucleotide”

-Authors should include negative controls results in all gels.

-Indicate in 3.1. that UGA is not a stop codon in mold mitochondrial genetic code, instead is a tryptophan codon.

-Add the reference to refer to the amino acid sequence alignment of mitoviruses RdRp.

-In 3.3. as I mentioned above, family Mitoviridae has only four genera. In the phylogenetic tree there is a fifth genus, Mitovirus, did the authors propose the creation of this genus, or what is the reference for this proposal? This point should be clarified since both new mitoviruses are included in this group. Indicate also that the tree is rooted.

-Explain how was done the semi quantitative PCR.

-Indicate the program used for secondary structure prediction.

-Since the authors have not identify in this work any isolate of the same MCG than GP3-1 free of mitoviruses, it is impossible to assign an effect of mitoviruses on fungal virulence. The results obtained do not indicate the relationship between hypovirulence of GP3-1 and the presence of mitoviruses, since this feature could depend on its genetic background. To demonstrate the effect of the mitoviruses infection on fungal virulence, authors should compare the same free- and infected fungal strain. BL1-1was already infected with both mitoviruses. Authors should show the gels with the result of the detection and quantification of both mitoviruses from GFP-3, BL1-1, BG5 and BG10. BL1-1 is the genetic background of BG5 and BG10? Bl1-1 strain was already infected with both mitoviruses, and the accumulation of both was very low, it is possible that the background of BG5 and BG10 is GFP-3? How the authors select one or the other strain if any of them have a marker? Additional experiments should be done to clarify all these points.

Comments on the Quality of English Language

Review English

Round 2

Reviewer 3 Report

Comments and Suggestions for Authors

The authors have addressed all my comments. My major concern was the possibility of having the genetic background of GP3-1 in BL1-1 and the authors have shown that BG5 and BG10 are indeed BL1-1.

Check carefully the text to avoid mistakes as: “Based on the PCR amplified with theTo understand the genome of fusion” (line 261) or “From PCR results, we found that the size of the amplified fragment of BG10 was the same to BL1-1, not to GP3-1. only affected by the genetic background of BL191, not GP3-1” (lines 265-267).

Change the word mitovirus in: 2.5. Determination of Mitocovirus in other A. rolfsii strains

In figure 3 legend change Metaviridae by Mitoviridae.

If possible, change the letter size of the phylogenetic tree, it is very small.

In my opinion this sentence it is not correct: “The UGA and UGG codon in the mitoviruses encoded tryptophan (Trp) rather than a stop codon”, it will be more correct to say: “The UGA (stop codon in the standard genetic code) and UGG codons in mitoviruses encoded tryptophan (Trp)”
